

# 1 2 Assessing and analysing the impact of land take pressures on agricultural land

Ece Aksoy¹*, Mirko Gregor², Christoph Schröder¹, Manuel Löhnertz², Geertrui Louwagie³
*¹ European Topic Centre on Urban, Land and Soil systems (ETC/ULS), University of Malaga, Malaga, Spain*
*² ETC/ULS, space4environment, Niederanven, Luxembourg.*
*³ European Environment Agency (EEA), Copenhagen, Denmark.*
*Correspondence to:* Ece Aksoy (*Ece.aksoy@uma.es*)
**Abstract .** Land, and here in particular soil, is a finite and essentially non-renewable resource. EU-wide, land
take, i.e. the increase of settlement area over time, consumes more than 1000 km² annually of which half is
actually sealed and, hence, lost under impermeable surfaces. Land take and in particular soil sealing has
already been identified as one of the major soil threats in the 2006 EC Communication "Towards a Thematic
Strategy on Soil Protection" and the Soil Thematic Strategy, and has been confirmed as such in the report on
the implementation of this strategy. The aim of this study is to relate the potential of land for a particular use
in a given region with the actual land use. This allows evaluating whether land (especially the soil dimension)
is used according to its (theoretical) potential. To this aim, the impact of several land cover flows related to
urban development on soils with a good, average and poor production potential were assessed and mapped.
Thus, the amount and quality (potential for agricultural production) of agricultural land lost between the years
2000 and 2006 was identified. In addition, areas with high productivity potential around urban areas,
indicating areas of potential future land use conflicts for Europe, were identified.

## 1 Introduction

Land use in Europe has changed drastically during the last fifty years, primarily in relation to the betterment
of human well-being and economic development, while unfortunately causing serious environmental problems
such as urban sprawl, soil sealing, loss of biodiversity, soil erosion, soil degradation, floods or desertification.
The changes in land use can also be interpreted as changes in the resources, services and goods which soils
offer to us; moreover, the type of land use change varies among different types of regions. Smith et al.
(2015) describe the effects of land use changes (increased change of agriculture to urban) on different
ecosystem services that are provided by soil decreased biomass and decreased availability of water for
agricultural use (provisioning services); decreased infiltration, storage, and soil-mediated water regulation
(regulating services); decreased genetic diversity (supporting service); and decreased natural environment
(cultural service).
Land use changes are a worldwide issue and the impacts of land use changes are the subject of several
studies. In recent years, several modelling and foresight studies of land use change have emerged with
European research projects, such as VOLANTE - Visions of Land Use Transitions in Europe (EU FP7 Project),
EU-LUPA- European Land Use Patterns (ESPON Project), SENSOR - Sustainable Impact Assessment on
Multifunctional Land Use in European Regions (EU FP6 Integrated Project) (Helming et al., 2006), EnviroGRIDS
(EU FP7 Project), ATEAM EU FP5 Project (Advanced Ecosystem Analysis and Modelling to search global climate
climate and land use change impacts on ecosystem vulnerability in Europe (Rounsevell et al., 2006),
EURURALIS Project – addressing socio-economic impacts associated with land use changes in the agricultural
sector (Klijn et al, 2005), SEAMLESS Project – approach for multi-scale modelling to asses sustainability
impacts of agricultural policies (van Ittersum et al., 2008), PRELUDE Project of EEA on scenarios for future
land use changes in Europe (Hoogeven & Ribeiro, 2007). Some examples from the literature on the impacts of
land-use change topic: Mancosu et al., (2015) develops different land-use change scenarios and discusses
their impacts on the Black sea region; Parras-Alcántara et al. (2013) examine the impacts of land use change
on soil carbon and nitrogen in a Mediterranean agricultural area; Cerda & Doerr (2007) investigate the
relations of different land use types with water use efficiency and soil conservation measures in dry
Mediterranean regions; Adugna & Abegaz (2016) discuss the effects of land use changes on the soil properties
in Ethiopia; Mohawesh et al. (2015) reveal the effects of land use changes on soil properties in Jordan and



results help in understanding the effects of land use changes on land degradation processes and carbon
sequestration potential and in formulating sound soil conservation plans; Wasak & Drewnik (2015) studied the
land use effects on soil organic carbon sequestration in calcareous Leptosols in the Tatra Mountains, Poland;
Muñoz-Rojas et al. (2015) analysed the long time series (1956-2007) impacts of land use and land cover
changes on organic carbon stocks in Mediterranean Soils; Liu et al. (2015) studied land use and climate
changes and their impacts on runoff in the Yarlung Zangbo river basin, China; Kalema et al. (2015) showed
the impacts of land use changes on woodlands in an Equatorial African Savanna; lastly, Trabaquini et al.
(2015) examined the effects of the land use changes of physical soil properties in the Brazilian savanna
environment.
Land take represents an increase of artificial surfaces or settlement areas (for e.g. residential, commercial,
industrial or infrastructure purposes) over time, usually at the expense of rural areas. This process can result
in an increase of scattered settlements in rural regions or in an expansion of urban areas around an urban
nucleus (urban sprawl, which is defined as "the physical pattern of low-density expansion of large urban areas,
under market conditions, mainly into the surrounding agricultural areas" (EEA, 2006)). A clear distinction is
usually difficult to make (Prokop et al, 2011).
Land take is a widespread phenomenon in Europe. The assessment as part of the EEA indicator "Land take"
(CSI 014/LSI 001) identifies extension of artificial land cover as one of the two major flows that consume
agricultural land; the other one is withdrawal of farming, which is supported by European policies (EEA, 2006).
Tóth (2012) analysed the impact of land take on the soil productivity using the JRC Cropland Productivity
Index map and combined it with CLC changes and socio-economic data. He concluded that the EU experiences
a constant decrease in production capacity (Tóth, 2012).
Soils are used to produce a range of biomass products that serve as food, feed, fibre and fuel. Biomass
production can be particularly relevant in biodiversity conservation and climate change mitigation efforts,
through supporting elements of green infrastructure and flood regulation (EEA, 2015). Biomass production is
one of the soil functions recognized in the European Union (CEC, 2006) and is severely affected by land take.
Urbanized land is not mainly used for agriculture, and furthermore, a large proportion of the land taken for
urbanization is actually sealed. Soil sealing can be considered as an almost irreversible process, since "de-
sealing" is very costly and the formation of new soil takes decades; i.e. 1cm in 100 years (Scheffer &
Schachtschabel, 2002). Accordingly, soil functions are commonly considered as lost when soils are covered
with impervious surfaces.
From the agricultural point of view, the land take is a soil/land loss for non-agricultural purposes, so that in a
way its effect is similar to soil degradation (caused by severe erosion), and might be considered as
complementary process. It is important to recognize, why it is of interest to compare different categories of
soil biomass productivity affected by land take and how these classes are connected to soil
erosion/degradation. Therefore, the aim of this study is to assess and analyze the impacts of several land
cover flows related to urban development (referred to as land take) between the years 2000 and 2006 on soils
with a good, average and poor biomass production potentials, and identify regions with major impact
(hotspots) in Europe.
**2 Material and methods**

**2.1 Material**

The main input data for this study is;
• Soil biomass productivity data on arable land (Toth et al., 2013)
• Land cover/use data (Corine Land Cover (CLC) 2000)
• Land cover/use changes (CLC changes and derived land cover flows (LCFs) between the years 2000-
95    2006).

The soil biomass productivity map on arable land was produced with the spatially explicit Soil Productivity
Model (SoilProd) for Europe by JRC (Toth et al. 2011) (Figure 1). This map provides composite cropland
Productivity Index scores, which are expressed on a scale from 1 to 10. Score 1 represents the lowest and 10
the highest biomass production potential. The Productivity Index is the sum of the Inherent Soil Productivity



Index and the Fertilizer Response Rate. The former results from an evaluation matrix set up for eight climatic zones, five inherent productivity classes (derived from second level taxonomic soil units), soil attribute information from the soil database (corrected for topographic conditions), and four available water capacity classes. The Fertilizer Response Rate takes account of the management practices applied. More details about the model and the map production process can be found in Tóth (2012) and Tóth et al. (2011).

**Figure 1.** Soil productivity data on arable land (pan-European grid layer) (JRC)

The soil biomass productivity data was provided by JRC, 1 km$^2$ raster data sets have full coverage of Europe but they are only valid for the corresponding land use types. Therefore, the appropriate CLC classes (based on CLC-Corilis 2000) were identified to build the masks for the extraction of the soil/land productivity layers. The used CLC classes are 2.1 'Arable land' (subclasses 211 'Non-irrigated arable land', 212 'Permanently irrigated land', and 213 'Rice fields') and 2.4 'Heterogeneous agricultural areas' (subclasses 241 'Annual crops associated with permanent crops' and 242 'Complex cultivation patterns') (Toth, 2012).

There are 9 major land cover flows (LCFs) on level 1 (Land and Ecosystem Accounting, LEAC, 2000-2006) (EEA, 2013) (Table 1). The combination of the "land take" flows LCF2 and LCF3 (urban residential sprawl and extension of economic sites and infrastructure) were used for this study. The impact calculation for Greece couldn't be done because of not having CLC 2006 and LCFs.

The technical assessment of land take on arable land is based on the land cover flows as described below:

- Definition LCF2: Urban residential sprawl: Land uptake by residential buildings altogether with associated services and urban infrastructure (classified in CLC 111 & 112) from non-artificial land (extension over sea may happen). Two sub-categories are distinguished, namely urban dense residential sprawl resulting in continuous urban fabric and urban diffuse residential sprawl resulting in discontinuous urban fabric.
- Definition LCF3: Sprawl of economic sites and infrastructures: Land uptake by new economic sites and infrastructures (including sport and leisure facilities) from non- artificial land (extension over sea may happen). This land cover flow includes eight sub-categories, namely sprawl of the following infrastructure on non-urban land; i.e. industrial & commercial sites, transport networks, harbors, airports, mines and quarries, dumpsites, construction, and sport and leisure facilities.(EEA, 2013)

## 2.2 Method

The schematic workflow of the study can be seen in Figure 2. Four main steps were followed to assess the impacts of land take pressures on agricultural lands analysis.

First of all, the soil biomass productivity data were classified into soils with a "good", "average" and "poor" capacity to provide biomass on arable land (step 1, Figure 2) with the aim of easier analysis, interpretation and calculation. This classification is performed based on the value distribution and their statistical parameters (mean and standard deviation). This means that the lower third of all values are classified as "poor" (class 1), the upper third as "good" (class 3), and the values in between as "average" (class 2).

Secondly, a mask was applied to the soil biomass productivity map (step 2, Figure 2) by using defined CLC classes according to the provisions of Tóth (2012). Then, after the classification and masking processes, the selected LCFs were overlaid onto the masked and classified data to extract the raster cells that contain a land cover change that is relevant for the analysis. This process in fact represents another masking process, as described in the Figure 2 (step 3). Lastly, the raster data were combined with the NUTS-3 reference units to compute the zonal statistics for each of the parameter combinations (impact of a particular LCF or combination of LCFs on a particular soil function potential (Figure 2, step 4).

**Figure 2.** Schematic workflow of the study



The final value of the impact of a particular LCF or combination of LCFs on the capacity of soils to supply a
particular soil function is expressed in relation to the share of that specific soil function potential in the NUTS-3
region. This means that the share of, e.g., good soils within a NUTS-3 region is the reference for the
calculation, not the entire area of the NUTS-3 region.
Moreover, for interpretation purposes the value ranges can be understood and verbally described regarding
their impact (expressed as percentages) as follows (ranked from very low to very high impact (green to red
colors in Figure 5)):
• very low impact;
• low impact;
• intermediate impact;
• high impact; and
• very high impact.
In addition, the descriptions of the outcomes make reference to relative and absolute impacts. Whereas
relative impacts correspond to the percentage values of the impact of a certain LCF on soils of a specific
capacity in a NUTS-3 region, the absolute impacts refer to the area (in hectare) that is affected by a particular
LCF. Depending on the size of the reference unit (that is, the area of soils of a specific capacity in a NUTS-3
region) high absolute values do not necessarily correspond to high relative values, while low absolute values
could well mean high relative values (when the total size of the reference area is very small).

**3 Results**

According to the results given in Table 2 and Figure 3, even though the highest share of the total arable lands
of the whole country coverage higher than 40% in Czech Republic, Germany, Denmark, Hungary, Lithuania,
Poland and RS; Turkey, France, Spain, Germany and Poland have over 15.000.000 ha arable lands in their
coverage respectively. Moreover, close to half (46.32%) of the arable lands in the whole study area and half of
the countries (18 out of 36 countries) have good productivity potentials. Over 80% of the arable lands in
Belgium (BE), Czech Republic (CZ), Denmark (DK), Ireland (IE), Sweden (SE) and United Kingdom (UK) have
good productivity potential. Over 80% of the arable lands in Bosnia and Herzegovina (BA), Spain (ES), Croatia
(HR), Lithuania (LT), Latvia (LV), Portugal (PT) and Kosovo (XK) have average productivity potential. Only one
country, Cyprus, has mostly poor biomass productivity potentials on its arable lands.

**Figure 3.** Graphic presentation of soil biomass productivity potentials on agricultural lands per country.

**Figure 4.** Distribution of soils in function of their potential for biomass production on arable land: proportions
of poor (left), average (center) and good (right) soils (in % of the total NUTS-3 region area); "less than 5 %"
means that the total area of arable land is smaller than 5 %. Note that the same colors might represent
different percentages as quantiles were used during the map production.

The distribution of the soils in function of their potential for biomass production on arable land per NUTS-3
area can be seen in Figure 4; the proportions are given in relation to the total area of each individual NUTS-3
region. By consequence, the maps nicely illustrate where poor, average or good soils dominate in Europe and
where they are only of minor importance.
Soils that are considered poor for biomass production on arable land mainly dominate in three European
regions, (i) Spain, (ii) central and north-eastern France, and (iii) south-eastern Europe (almost entire Turkey



and large parts of Greece). Almost all other regions have an intermediate to low share of poor soils for the
provision of biomass on arable land. Of the first 20 NUTS-3 regions across Europe fourteen are located in
Turkey (Figure 4). The others are located in the UK, France and Cyprus (NUTS-3 region boundary corresponds
to the entire country). However, most of the mentioned regions show very low to intermediate impact of
urban expansion; Cyprus shows a high impact though.
Average soils for arable biomass provision are widespread across Europe and can be found in large parts of
Spain and Italy, Hungary, Poland and the southern Baltic countries (Lithuania and Latvia), as well as in
regions of Germany, France, Bulgaria and Greece possess average soils. Low shares of average soils can be
found in Turkey, parts of Greece, Bulgaria and Romania, the Czech Republic, parts of Germany and France,
the UK and Scandinavia. The number of NUTS-3 regions with a high to very high share of average soils for
biomass provision on arable land (Figure 3) is substantially higher compared to those with a high share of
poor soils. 32 regions have a majority share, that is, of more than 50 %, in the respective NUTS-3 region
(only one NUTS-3 region for poor soils), with the highest values of over 70 % in one Spanish (ES418,
Valladolid) and two Italian regions (ITH36 and ITH57, Padova and Ravenna, respectively). In general, there is
a high share of Italian regions within those 32 regions (12 NUTS-3 regions), often located in or close to the Po
Valley which used to be one of the most fertile areas in Europe; another remarkable hotspot is Lithuania with
5 regions.
Good soils for the provision of biomass on arable land dominate in large parts of north-western Europe, such
as lots of regions in the UK, north-western France, the Benelux countries, Germany, Denmark, Poland, Czech
Republic, Hungary, and Bulgaria. Even some regions in central Turkey have a high share of good soils. Low
shares can be found mainly in the Western Balkan countries, the Iberian Peninsula, Romania, the Baltic
countries and some regions in Finland and Sweden. Compared to the average soils, the number of NUTS-3
regions with a very high share of good soils is even bigger; almost 140 regions have a majority share of good
soils, with the upper seven regions exceeding 80 % (four regions in the UK, two in Romania and one in
Germany) (Figure 4).
The highest land take impact on the biomass productivity potentials of arable land was found in Albania (AL)
(3.97%), the Netherlands (NL) (1.45%), Cyprus (CY) (1.39%) and Ireland (IE) (0.76) (Table 3 and Figure 6).
However, when expressing the impacts on an absolute (in hectare) rather than on a relative (in percentage)
basis, Spain, France and Germany rank highest (with 71338 ha, 52096 ha and 47620 ha, respectively). Thus,
even though the relative impact may be low in some countries, the absolute impact may be quite high. For
example, while the share of land with good and average productivity potential is very similar (0.5% and
0.44% respectively), the total area of land with good productivity potential is far lower (4 341 ha) than that of
average productivity potential (59 786 ha). (Table 3). Therefore, it is better to consider the absolute and
relative values in parallel.
Figure 5 describes the impact of land take (the combination of LCF2 and LCF3, i.e. residential, commercial,
industrial and infrastructure-related extension) on arable land with a poor, average and good potential for the
provision of biomass.

**Figure 5.** Percentage decline (per NUTS 3 area) of arable land area with poor (left), average (centre) and
good (right) production potential due to urban residential, commercial, industrial and infrastructure-related
extension (LCF2 and LCF3) between 2000 and 2006; "less than 5 %" means that the total area of arable land
is smaller than 5%

**Figure 6.** Graphic presentation of land take impact on agricultural lands per country

In general, the map illustrates that regions with a very high impact of urban land take on poor soils are
scattered across Europe; there is no geographic area with a striking clustering of such regions. However, the
south-eastern part of Europe only contains a few NUTS-3 regions with a high impact: Cyprus, Istanbul, and
one region in Romania (Galati, RO224). Also, Albania possesses some regions with a high to very high impact
of urban land take. On the other hand, the NUTS-3 regions with the highest relative impact still possess only
low to very low share of poor soils within the NUTS-3 regions. Most of these regions are located in north-



western Europe (UK, Ireland, Germany), some isolated ones can be found in south-western France, Italy and
Poland. When looking at absolute impacts (in terms of total area affected) of urban expansion on poor arable
soils, four regions stand out, located in southern Europe. Except for Seville (ES618), all other regions (Cyprus,
Istanbul and Valladolid) also have high to very high relative impacts.
Figure 5 clearly shows that on the one hand regions with a high to very high impact of urban expansion
activities on average soils are distributed across Europe, but that on the other hand some clusters exist. Most
striking is Albania that comprises the two regions with the highest relative impact (AL00B and AL002, Tirana
and Durres, with 14.8 and 12.1 %, resp.); followed by the Netherlands, Germany, Italy and Spain that also
possess a number of regions with a very high impact of urban land take on average soils.
In terms of absolute values, only a few of the previous regions show up on the leader board. Interestingly,
both Albanian regions also possess a large absolute value (2 752 and 1 997 ha, resp.). But the region with by
far the highest absolute value is the region of Madrid (ES300) with 11 854 ha of average soils lost due to land
take, which corresponds to 5.2 % in relative terms. The absolute value of Madrid is more than double of that
of the second highest region which is another Spanish region (Toledo), followed by two other Spanish regions
(Ciudad Real and Zaragoza). Remarkably, many more Spanish regions follow amongst the regions next in
order. This implies a very high absolute loss of average soils due to land take, but with often less relevance
when it comes to the relative impact (often intermediate, sometimes high values regarding the share of soils
with average potential in a particular NUTS-3 area).
Regarding the distribution of regions with a very high impact of land take on good soils, some clusters of
regions/hotspots exist. One is located in the Netherlands and western Germany, another one in the Western
Balkans (including Albania), a third one from northern Italy (Umbria and Po valley) to south-eastern France
(Alpes and Provence, Rhone Valley), a fourth one on the Iberian Peninsula, and a last one in Ireland. The
relative impact ranges from 38.9 % in Tirana (AL00B) over 34.7 % (NL332, Agglomeratie 's-Gravenhage),
27.7 % (NL327, Het Gooi en Vechtstreek) and 15 % (NL325, Zaanstreek) to several regions between 10.6 and
5 % impact.
In terms of absolute values, most of those regions with very high relative impact values do not score very
high, though. Only two Albanian regions as well as one Irish region stand out. Other than that, there are five
other regions (next to Tirana, AL00B) that have more than 2 000 ha of impacted good soils on arable land.
Three of those regions are located in France, one in Turkey and one in the Czech Republic. In terms of relative
impacts, they possess intermediate to high values (between 0.34 % and 0.81 %). Interestingly, many of the
high-ranked regions possess a share of more than 50 % of good soils; however, there are also some regions
with a very low share. One of those regions is again Tirana with a share of 3.4 %, others are AL00A (Shkoder,
5.5 %) and ES523 (Valencia, 4.3 %). The latter two also show very high relative values of the impact of land
take on the good soils, that is, of the limited area with good soils available, a high share is affected by land
take.

## 4 Discussion


In general, most of the arable lands have good productivity potentials, both at country level (18 Countries out
of 36) and when considering the entire coverage of the study area (46.32%).
However, the European picture is, as expected, very heterogeneous. The urban residential expansion and
extension of economic sites and infrastructure activities is spatially distributed across Europe, with very low
(green) to very high (red) impact on the biomass productivity of arable land. Several hotspot areas can be
identified in which land take clearly affects soils with a capacity to provide biomass.
The highest share of arable land affected by land take was found in Albania (AL) (3.97%), the Netherlands
(NL) (1.45%), Cyprus (CY) (1.39%) and Ireland (IE) (0.76). However, when the impacted lands are
considered in hectare, Spain, France and Germany are on top of the leaderboard. High and very high impacts
on good land can mainly be detected in regions in Ireland, Spain, France, Germany, Italy and the Balkan
countries. Average land is strongly impacted in Albania, the Netherlands, Germany, Italy and Italy. Very high
impacts of urban land take on poor soils are scattered across Europe.



When taking the gross domestic product of the outstanding regions into account, there seems to be no direct relation to the economic situation of a region. Both well-developed and less-developed regions experience high to very high impacts of land take-related land cover flows on the soil productivity.

Several hotspot areas are identified in which land take clearly affects soils with a capacity to provide biomass. The Madrid region is one of the hotspots of urban development in Europe, experiencing a rate of 50 % growth in the 1990s, compared to 25 % national and 5,4 % EU average rates (EEA, 2006). The trend attenuated between 2000 and 2006 (around 20 %), but is still present. According to Díaz-Pacheco & García-Palomares (2014) the urban land surface grew at a rate in excess of 4 % per year. Tóth (2012) shows that the urban sprawl of Madrid occurred to a large extent on agricultural land. According to the EEA Report (EEA, 2006) major drivers are (i) the growing demand for first and second homes caused by economic growth and low interest rates despite a rather modest population growth; (ii) increased mobility; (iii) increasing housing prices, which force more people to move further and further into the city's hinterland; and (iv) a weak planning framework. The reasons for land take differ from country to country; nevertheless, these major drivers which were given for Madrid region might be valid for most of the regions or countries in Europe with the addition of some items such as ; new developments along transportation axes, tourism and coastline diffusion in general. Moreover, also the OECD reports (OECD, 2007) about rapid and partly unplanned development that, amongst other, led to urban sprawl in the Madrid region.

Alongside the situation in the Madrid region, the EEA (EEA, 2006) also presents the example of the occurring urban sprawl along the Spanish and Portuguese coastlines. In these areas, sprawl mainly consists of diffuse settlements adjacent to or disconnected from concentrated urban centres. This residential sprawl is responsible for more than 45% of coastal zone land transformation into artificial surfaces. In Portugal, 50% of the urban areas are located between Lisbon/Setubal and Porto/Viana do Castelo within 13 km from the shoreline, hence covering only 13 % of the total land area. In Spain economic growth, legislative flexibility and tourism resulted in an increased number of households and second homes along the coast, in combination with infrastructure and leisure facilities development.

Outside the Iberian Peninsula, the Po Valley and the adjacent Emilia-Romagna Plain (ERP) have a long history of urban expansion. The valley has soils that are amongst the most fertile in Europe. Even though the entire region is called "Food Valley", more and more of its agricultural area is irreversibly converted into urban fabric, either for residential, or industrial and commercial use, continuing at a rate of 1 ha per day (EC, 2011). The movie "Il suolo minacciato" ("Land under threat") presented during the Green Week 2011 uses the example of these two confronting pressures on land to highlight what is currently happening in this region. Malucelli et al. (2014) confirm that while the extent of woodland, grassland, natural areas and wetlands in the ERP did not change significantly, urban and industrial areas increased to the detriment almost exclusively of cropland. The analysis in the current study highlights that mainly good and average land is affected.

The impacts of land take on regions in southern France are also already described and explained in the EEA report on urban sprawl (EEA, 2006). This so-called "inverse T" of urban sprawl along the Rhone valley down to the Mediterranean coast is caused by new developments along transportation axes and coastlines (which is often connected to river valleys).

Another prominent and well-known region of urban sprawl and related land take is the Dublin metropolitan area, which can be recognised on the maps of average and good soils. In the past, population growth and economic development were responsible for the expansion of the metropolitan area further to the outskirts. (EEA, 2006)

In Germany, land take is most prominent in the region comprising the 'Ruhrgebiet' (in particular the regions around its core), in parts of southern Germany, but also in eastern Germany, particularly in some regions which are experiencing an improvement of their economic situation (e.g. Leipzig). Prokop et al. (2011) state that despite having defined a target of reducing land take to 30ha/day until 2020, the measures taken so far have not been sufficient.

In the Netherlands most regions have experienced and are still experiencing rapid urban expansion along the urban-rural fringes during the past decades; which is still on-going although spatial planning policies were seeking to promote compact urban developments (Nabielek et al., 2013). This increase in land take is also documented in Prokop et al. (2011), showing the constant increase of built-up area between the 1960s and 2006 (Fig. 49 in Prokop et al., 2011). A similar picture appears in the Flanders region (Belgium) where the




typical ribbon development continues with a rate of 6 ha per day of which 5 ha is due to residential sprawl
(Gregor et al., 2015).
Regarding the conversion of arable land to urbanised areas in the central and eastern European countries, it
can be assumed that the accession to the EU in 2004 and the related economic development together with
benefits from Regional Development programmes were the leading driving forces to the expansion of
residential, but mainly industrial and commercial areas, primarily at the expense of good and average land.
Very recent statistics on the cohesion funding amount allocated per member state (EC, 2015) confirm that
some of the eastern European countries rank amongst the top; e.g. Poland is the country with the highest
amount allocated, while the Czech Republic and Hungary rank fourth and sixth, respectively.
Without being a member state of the EU, Albania has undergone significant changes with regards to urban
expansion and land take. In particular average and good soils for providing biomass on arable land have been
converted into artificial surfaces, according to the most recent assessment of the EEA CSI 014 on land take.
This has happened at the expense of grassland and mixed farmland (in total 73 % of the total land uptake)
which is of relevance in this context of arable land. Likewise, also in Bosnia and Herzegovina 72 % of the total
land take occurred on grassland or mixed farmland areas.

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

**Figures**
**Figure    1.**    Soil    productivity    data    on    arable    land    (pan-European    grid    layer)    (JRC)

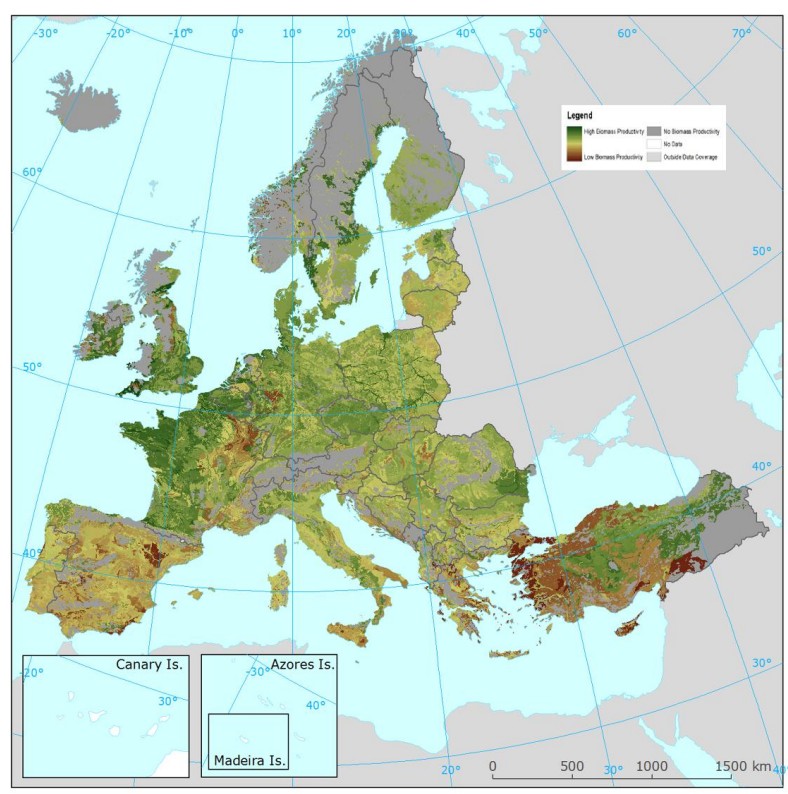








**Figure 2.** Schematic workflow of the study



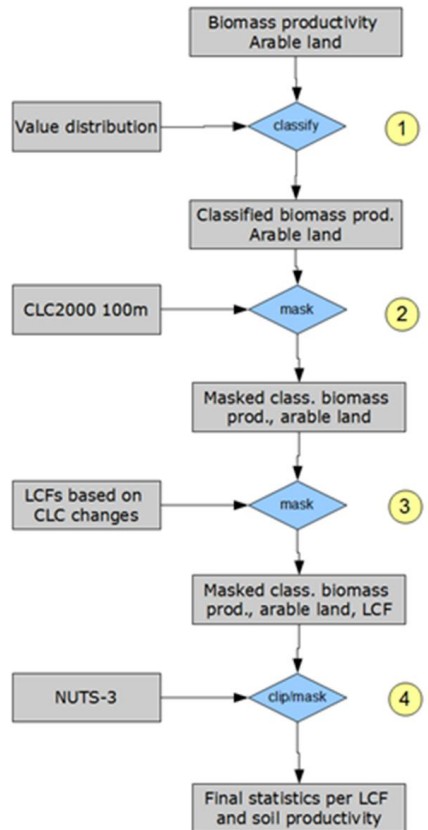


**Figure 3.** Graphic presentation of soil biomass productivity potentials on agricultural lands per country.

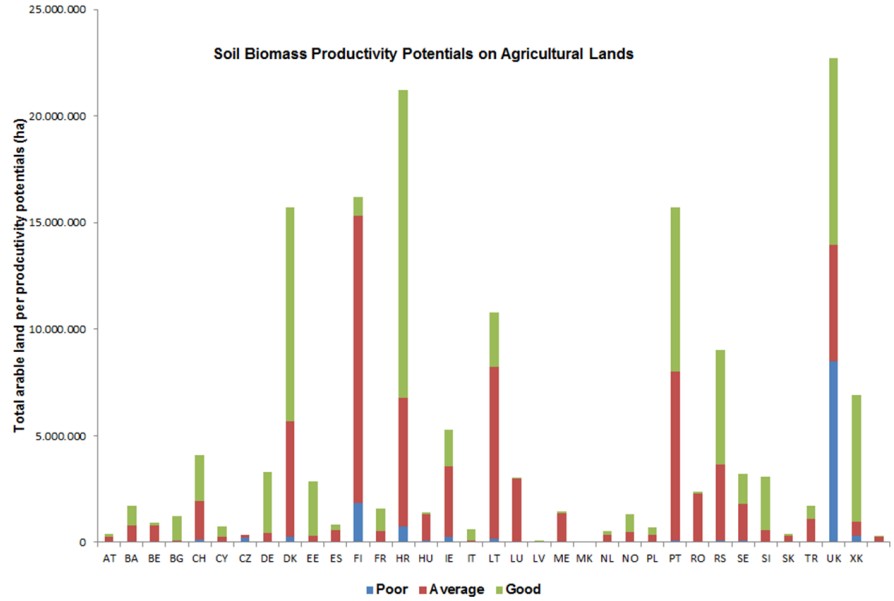




**Figure 4.** Distribution of soils in function of their potential for biomass production on arable land: proportions of poor (left), average (center) and good (right) soils (in % of the total NUTS-3 region area); "less than 5 %" means that the total area of arable land is smaller than 5 %. Note that the same colors might represent different percentages as quantiles were used during the map production.

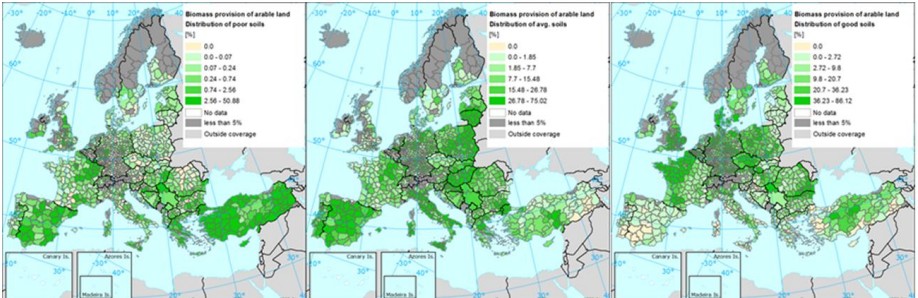

**Figure 5.** Percentage decline (per NUTS 3 area) of arable land area with poor (left), average (centre) and good (right) production potential due to urban residential, commercial, industrial and infrastructure-related extension (LCF2 and LCF3) between 2000 and 2006; "less than 5 %" means that the total area of arable land is smaller than 5%

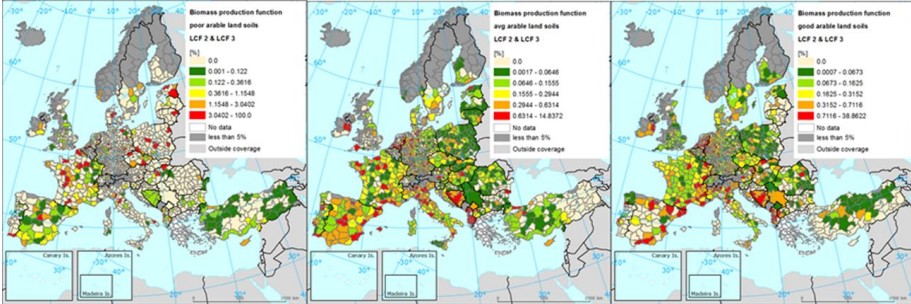

**Figure 6.** Graphic presentation of land take impact on agricultural lands per country

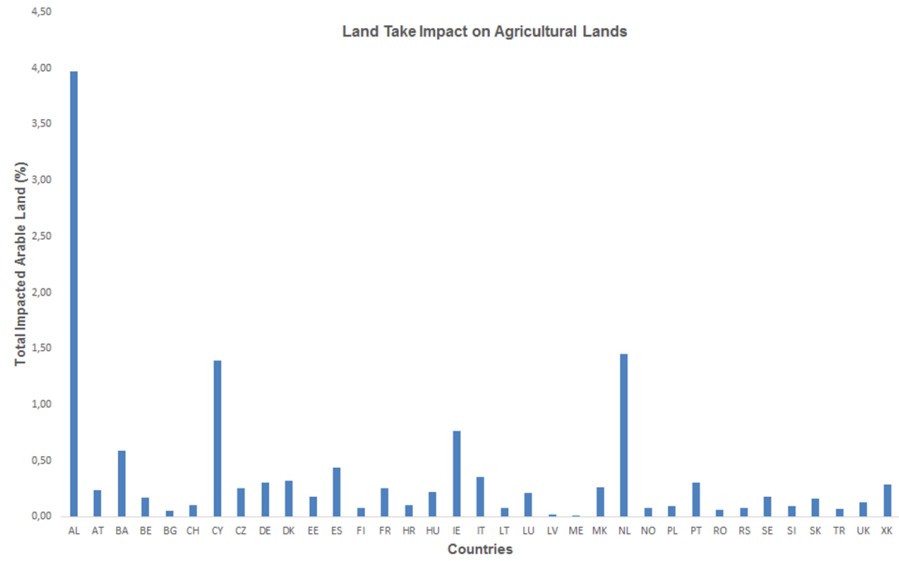





**Tables**
**Table 1.** Major land cover flows (LCFs) on Level 1 (EEA, 2013)

| Code | Major Type of Cover change |
|------|----------------------------|
| LCF1 | Urban land management |
| LCF2 | Urban residential sprawl |
| LCF3 | Extension of economic sites and infrastructure |
| LCF4 | Agriculture internal conversions |
| LCF5 | Conversion from forested and natural land to agriculture |
| LCF6 | Withdrawal of farming |
| LCF7 | Forests creation and management |
| LCF8 | Water body creation and management |
| LCF9 | Changes of land cover due to natural and multiple causes |


**Table 2.** Statistical distribution of arable lands according to their biomass production potential per country
(green color shows the major share) (Abbreviations of the Countries; AL- Albania, AT – Austria, BA-Bosnia and
Herzegovina, BE-Belgium, BG-Bulgaria, CH-Switzerland, CY-Cyprus, CZ-Czech Republic, DE-Germany, DK-
Denmark, EE-Estonia, EL-Greece, ES-Spain, FI-Finland, FR-France, HR-Croatia, HU-Hungary, IE-Ireland, IT-
Italy, LT-Lithuania, LU-Luxemburg, LV-Latvia, ME-Montenegro, MK- Macedonia, NL-the Netherlands, NO-
Norway, PL-Poland, PT-Portugal, RO-Romania, RS-Serbia, SE-Sweden, SI-Slovenia, SK-Slovakia, TR-Turkey,
UK-United Kingdom, XK-Kosova)

| Country | Country [Km2] | Total Arable Land [Km2] | Arable Land Proportion [%] | Soil Biomass Productivity Potential per Total Arable Land | | |
|---------|---------------|-------------------------|----------------------------|------|------|------|
| | | | | Poor [%] | Average [%] | Good [%] |
| AL | 28755.06 | 3729.06 | 12.97 | 6.99 | 65.48 | 27.53 |
| AT | 83947.82 | 17291.73 | 20.6 | 0.93 | 43.96 | 55.11 |
| BA | 51399.37 | 9032.91 | 17.57 | 3.2 | 84.69 | 12.11 |
| BE | 30664.19 | 12123.74 | 39.54 | 0.25 | 7.32 | 92.43 |
| BG | 110988.76 | 41120.23 | 37.05 | 2.81 | 44.65 | 52.54 |
| CH | 41287.33 | 7605.61 | 18.42 | 1.31 | 31.71 | 66.98 |
| CY | 9249.11 | 3468.26 | 37.5 | 62 | 38 | 0 |
| CZ | 78869.52 | 33054.4 | 41.91 | 0.19 | 13.15 | 86.66 |
| DE | 357737.29 | 157211.86 | 43.95 | 1.58 | 34.41 | 64.01 |
| DK | 43174.76 | 28487.19 | 65.98 | 0.09 | 11.33 | 88.58 |
| EE | 45335.44 | 8338.46 | 18.39 | 1.39 | 65.76 | 32.84 |
| EL | 131735.85 | 28320.05 | 21.5 | 12.47 | 74.54 | 12.99 |
| ES | 505980.28 | 161978.31 | 32.01 | 11.51 | 83.08 | 5.41 |
| FI | 337616.92 | 15956.63 | 4.73 | 0.7 | 32.69 | 66.61 |
| FR | 638480.71 | 212195.3 | 33.23 | 3.58 | 28.45 | 67.97 |
| HR | 56599.65 | 13861.69 | 24.49 | 4.85 | 91.22 | 3.93 |
| HU | 93012.99 | 52795.47 | 56.76 | 4.9 | 62.96 | 32.13 |
| IE | 69956.69 | 6336.24 | 9.06 | 7.42 | 9.04 | 83.54 |
| IT | 300620.28 | 107714.14 | 35.83 | 1.78 | 74.5 | 23.72 |
| LT | 64901.2 | 30396.19 | 46.83 | 0.09 | 98.45 | 1.46 |
| LU | 2595.06 | 847.7 | 32.67 | 13.37 | 38.21 | 48.41 |
| LV | 64596.24 | 14582.72 | 22.58 | 1.26 | 91.14 | 7.6 |
| ME | 13878.81 | 166.5 | 1.2 | 20.16 | 32.64 | 47.2 |



| | | | | | | |
|---|---|---|---|---|---|---|
| **MK** | 25436.12 | 5191.04 | 20.41 | 3.9 | 64.16 | 31.95 |
| **NL** | 37373.99 | 13059.17 | 34.94 | 1.25 | 36.62 | 62.13 |
| **NO** | 323024.51 | 7033.67 | 2.18 | 3.97 | 46.07 | 49.97 |
| **PL** | 311942.39 | 157056.55 | 50.35 | 0.47 | 50.49 | 49.04 |
| **PT** | 91969.54 | 23630.11 | 25.69 | 2.33 | 94.53 | 3.14 |
| **RO** | 238364.06 | 90033.96 | 37.77 | 1.12 | 39.45 | 59.43 |
| **RS** | 77313.57 | 32165.26 | 41.6 | 3.06 | 53.01 | 43.93 |
| **SE** | 449563.7 | 30977.11 | 6.89 | 0.83 | 17.69 | 81.48 |
| **SI** | 20273.58 | 3900.67 | 19.24 | 2.41 | 79.7 | 17.9 |
| **SK** | 49027.63 | 17085.84 | 34.85 | 0.38 | 64.04 | 35.58 |
| **TR** | 780290.77 | 226986.37 | 29.09 | 37.49 | 23.93 | 38.58 |
| **UK** | 244619.49 | 68939.64 | 28.18 | 4.68 | 9.15 | 86.17 |
| **XK** | 11004.64 | 2877.41 | 26.15 | 0.99 | 94.53 | 4.48 |
| **Grand Total** | 5821587.32 | 1645551.2 | 28.27 | 8.23 | 45.45 | 46.32 |


**Table 3.** Statistical distribution of the land take impact on arable lands per country between 2000-2006 years
(Abbreviations of the Countries; AL- Albania, AT – Austria, BA-Bosnia and Herzegovina, BE-Belgium, BG-
Bulgaria, CH-Switzerland, CY-Cyprus, CZ-Czech Republic, DE-Germany, DK-Denmark, EE-Estonia, ES-Spain,
FI-Finland, FR-France, HR-Croatia, HU-Hungary, IE-Ireland, IT-Italy, LT-Lithuania, LU-Luxemburg, LV-Latvia,
ME-Montenegro, MK- Macedonia, NL-the Netherlands, NO-Norway, PL-Poland, PT-Portugal, RO-Romania, RS-
Serbia, SE-Sweden, SI-Slovenia, SK-Slovakia, TR-Turkey, UK-United Kingdom, XK-Kosova)

| Country | Total Arable Land (ha) | Total Impact on Arable Land (ha) | Impact On Arable Land (ha) | | | Total Impacted Arable Land (%) | Impact On Total Arable Land (%) | | |
|---|---|---|---|---|---|---|---|---|---|
| | | | **Poor** | **Avg** | **Good** | | **Poor** | **Avg** | **Good** |
| AL | 372906 | 14795 | 539 | 8672 | 5584 | 3.97 | 2.07 | 3.55 | 5.44 |
| AT | 1729173 | 4137 | 20 | 1478 | 2639 | 0.24 | 0.12 | 0.19 | 0.28 |
| BA | 903291 | 5329 | 61 | 4100 | 1168 | 0.59 | 0.21 | 0.54 | 1.07 |
| BE | 1212374 | 2027 | 7 | 111 | 1909 | 0.17 | 0.23 | 0.13 | 0.17 |
| BG | 4112023 | 1920 | 145 | 1289 | 486 | 0.05 | 0.13 | 0.07 | 0.02 |
| CH | 760561 | 784 | 12 | 157 | 615 | 0.10 | 0.12 | 0.07 | 0.12 |
| CY | 346826 | 4816 | 4087 | 729 | 0 | 1.39 | 1.90 | 0.55 | 0.00 |
| CZ | 3305440 | 8390 | 103 | 900 | 7387 | 0.25 | 1.64 | 0.21 | 0.26 |
| DE | 15721186 | 47620 | 1605 | 16053 | 29962 | 0.30 | 0.65 | 0.30 | 0.30 |
| DK | 2848719 | 9250 | 18 | 1001 | 8231 | 0.32 | 0.69 | 0.31 | 0.33 |
| EE | 833846 | 1522 | 491 | 929 | 102 | 0.18 | 4.23 | 0.17 | 0.04 |
| ES | 16197831 | 71338 | 7211 | 59786 | 4341 | 0.44 | 0.39 | 0.44 | 0.50 |
| FI | 1595663 | 1207 | 0 | 246 | 961 | 0.08 | 0.00 | 0.05 | 0.09 |
| FR | 21219530 | 52096 | 2919 | 12376 | 36801 | 0.25 | 0.38 | 0.20 | 0.26 |
| HR | 1386169 | 1409 | 0 | 1389 | 20 | 0.10 | 0.00 | 0.11 | 0.04 |
| HU | 5279547 | 11382 | 374 | 7469 | 3539 | 0.22 | 0.14 | 0.22 | 0.21 |
| IE | 633624 | 4806 | 193 | 765 | 3848 | 0.76 | 0.41 | 1.34 | 0.73 |
| IT | 10771414 | 37484 | 179 | 26747 | 10558 | 0.35 | 0.09 | 0.33 | 0.41 |
| LT | 3039619 | 2522 | 17 | 2472 | 33 | 0.08 | 0.64 | 0.08 | 0.07 |





| | | | | | | | | |
|---|---|---|---|---|---|---|---|---|
| LU | 84770 | 177 | 75 | 37 | 65 | 0.21 | 0.66 0.11 | 0.16 |
| LV | 1458272 | 316 | 42 | 243 | 31 | 0.02 | 0.23 0.02 | 0.03 |
| ME | 16650 | 1 | 0 | 1 | 0 | 0.01 | 0.00 0.02 | 0.00 |
| MK | 519104 | 1330 | 6 | 712 | 612 | 0.26 | 0.03 0.21 | 0.37 |
| NL | 1305917 | 18874 | 213 | 6943 | 11718 | 1.45 | 1.30 1.45 | 1.44 |
| NO | 703367 | 557 | 20 | 244 | 293 | 0.08 | 0.07 0.08 | 0.08 |
| PL | 15705655 | 14246 | 622 | 6629 | 6995 | 0.09 | 0.85 0.08 | 0.09 |
| PT | 2363011 | 7099 | 79 | 6840 | 180 | 0.30 | 0.14 0.31 | 0.24 |
| RO | 9003396 | 5828 | 59 | 2178 | 3591 | 0.06 | 0.06 0.06 | 0.07 |
| RS | 3216526 | 2430 | 0 | 792 | 1638 | 0.08 | 0.00 0.05 | 0.12 |
| SE | 3097711 | 5728 | 99 | 734 | 4895 | 0.18 | 0.38 0.13 | 0.19 |
| SI | 390067 | 332 | 11 | 280 | 41 | 0.09 | 0.12 0.09 | 0.06 |
| SK | 1708584 | 2660 | 0 | 1445 | 1215 | 0.16 | 0.00 0.13 | 0.20 |
| TR | 22698637 | 16761 | 7153 | 4259 | 5349 | 0.07 | 0.08 0.08 | 0.06 |
| UK | 6893964 | 8832 | 671 | 1552 | 6609 | 0.13 | 0.21 0.25 | 0.11 |
| XK | 287741 | 840 | 0 | 832 | 8 | 0.29 | 0.00 0.31 | 0.06 |
| Total | 161723114 | 368845 | 27031 | 180390 | 161424 | 0.23 | 0.2 0.25 | 0.21 |
