# Peer review of "2 Assessing and analysing the impact of land take pressures on agricultural land"

_Solid Earth, 2016_

## Referee Comment (RC1) · Anonymous Referee #1 · 3 Jan 2017

Theme of the manuscript is in the scope of the journal and is of interest to the scientific community. Methodological approach of the study is adequate and - with a few exception - is clearly presented. The presentation of land cover flows (Table 1) should be clarified by explaining the departure land cover class (arable) as well.

Results are mostly new findings, discussed in comparison to previous studies on the field. In certain sections the comparative text is a bit wordy; simpler and clearer discussion is preferred. (eg. "The impacts of land take on regions in southern France are also already described and explained in the EEA report on urban sprawl (EEA, 2006)").

Some of the findings are not supported by facts. For example land take in Central and Eastern European countries is attributed to the EU accession of these countries. To my knowledge, land take was already a very significant process in these countries before

joining the EU. Please check and verify.

Citations in the text and the reference list should be revised as well. (eg. 2006 EC Communication "Towards a Thematic Strategy on Soil Protection" vs. Soil Thematic Strategy; CSI 014/LSI 001; Green Week 2011).

Figures are informative and mostly OK, but there are also errors. Country abbreviations on figure 3 are switched one space to the right. Please correct. Maps on Figure 4 and 6 are too small. Consider presenting these in large maps. (As the journal is primarily uses on-line distribution, this should not be a problem. Please check it with he editors.)

With these minor revision needs I suggest the paper to be accepted for publication.

---

## Short Comment (SC1) · 16 Jan 2017

Dear reviewer, Thank you very much for your valuable comments and suggestions. We'll consider all of them. Regards, Ece Aksoy
* * *

---

## Referee Comment (RC2) · Anonymous Referee #2 · 1 Feb 2017

**General comments**

The authors present an analysis of land take in arable land in Europe between 2000 and 2006. The novelty of the study is that it is done according to potential biomass productivity levels. The methodology and results are sound but their reporting could be improved. Also, an overall conclusion should be added.

The authors sometimes refer to agricultural soils, sometimes to arable land, apparently equalling both. This should be clarified. In the captions of figures 3 and 6, "arable" should be used instead of "agricultural".

**Specific comments**

*L137:* Not clear whether the classification into 'poor', 'average' and 'good' is based on mean and standard deviation or on 33 and 66 percentiles. If it is the latter, please

[Figure]

remove reference to mean and standard deviation.

*Figure 4:* the colour ramp should be the same for the three sub-figures rather than quantile based. The figures should also be of better resolution/bigger.

*Figure 5:* same comments as figure 4.
* * *

---

## Short Comment (SC2) · 7 Feb 2017

Dear reviewer, Thank you very much for your valuable comments and suggestions. We'll consider all of them in our revised paper. Regards, Ece Aksoy
* * *

---

## Author Comment (AC1) · 3 Mar 2017

Theme of the manuscript is in the scope of the journal and is of interest to the scientific community. Methodological approach of the study is adequate and - with a few exception - is clearly presented. Author's response 1: Thank you very much for your comments and valuable suggestions.

The presentation of land cover flows (Table 1) should be clarified by explaining the departure land cover class (arable) as well. Author's response 2: Unfortunately, I couldn't understand well the meaning of "departure land cover class (arable)". The list of the land cover flows that was given in Table 1 is the major LCFs, level 1 classes and they are calculated for each of the land covers per pixel and given as a huge matrix as you

know. We only used LCF 2 and 3 classes as "land take" and analyzed the impacts of those LCFs on arable land. Some details of the table is explained in page 3 and more detail can be find in Land and Ecosystem Accounting, LEAC, EEA, 2013.

Results are mostly new findings, discussed in comparison to previous studies on the field. In certain sections the comparative text is a bit wordy; simpler and clearer discussion is preferred. (eg. "The impacts of land take on regions in southern France are also already described and explained in the EEA report on urban sprawl (EEA, 2006)"). Author's response 3: Thanks for the comment. You are right, it might be long but includes necessary information to describe hot-spots and country based details. We improved this section by adding more comparisons and descriptions of countries in detail as a main output of the study, which was also suggested by former referees.

Some of the findings are not supported by facts. For example land take in Central and Eastern European countries is attributed to the EU accession of these countries. To my knowledge, land take was already a very significant process in these countries before joining the EU. Please check and verify. Author's response 4: The analysing period of this study is 2000-2006 and reflects the high proportion of land take on agriculture land for this period (before and after accession) in Figure 5 with the red color, visible as hot-spot.

Citations in the text and the reference list should be revised as well. (eg. 2006 EC Communication "Towards a Thematic Strategy on Soil Protection" vs. Soil Thematic Strategy; CSI 014/LSI 001; Green Week 2011). Author's response 5: Thank you for the comment. I tried to revise the references in the manuscript as you suggested.

Figures are informative and mostly OK, but there are also errors. Country abbreviations on figure 3 are switched one space to the right. Please correct. Author's response 6: Thank you for this very careful and important comment. Corrected. Maps on Figure 4 and 6 are too small. Consider presenting these in large maps. (As the journal is primarily uses on-line distribution, this should not be a problem. Please check it with

the editors.) Author's response 7: Yes, this is the problem of trying to map very small numbers, that's why we gave the impacts in NUTS3, otherwise, they were not visible. The original of the image is bigger and more visible but on the paper, it looks small. If it's possible to publish them bigger, I'll do it by consulting the editor. Thanks for this comment.

With these minor revision needs I suggest the paper to be accepted for publication.

Please also note the supplement to this comment:
http://www.solid-earth-discuss.net/se-2016-154/se-2016-154-AC1-supplement.pdf

---

## Author Comment (AC2) · 3 Mar 2017

General comments The authors present an analysis of land take in arable land in Europe between 2000 and 2006. The novelty of the study is that it is done according to potential biomass productivity levels. The methodology and results are sound but their reporting could be improved. Author's response 1: Thank you very much for your comments and valuable suggestions.

Also, an overall conclusion should be added. Author's response 2: Right, conclusion will be added to the manuscript.

The authors sometimes refer to agricultural soils, sometimes to arable land, apparently

equalling both. This should be clarified. Author's response 3: You're right, thanks for this important suggestion. Most of the agricultural land including title is changed. But some of them were kept because of the correct meaning of the agriculture and also the references.

In the captions of figures 3 and 6, "arable" should be used instead of "agricultural". Author's response 4: Thank you for this very careful and important comment. Corrected.

Specific comments

L137: Not clear whether the classification into 'poor', 'average' and 'good' is based on mean and standard deviation or on 33 and 66 percentiles. If it is the latter, please remove reference to mean and standard deviation. Author's response 5: Right, corrected (mean and standard deviation are deleted).

Figure 4: the colour ramp should be the same for the three sub-figures rather than quantile based. The figures should also be of better resolution/bigger. Figure 5: same comments as figure 4. Author's response 6: This issue had been discussed with the other colleagues as well. The problem is that trying to map very small numbers, that's why we gave the impacts in NUTS3, otherwise, they were not visible. If we didn't change the color ramp for the each of the sub-figures, some of the sub-figures would have only one color. Therefore, this is the only way to visualize those different ranges of values. The original of the image is much bigger and more visible because of the better resolution but since they're embedded to the manuscript with the lower resolution for review purpose, they look small and bad quality. I'll upload each of the figures separately in good quality, so I hope it'll be much better in the published version.

Please also note the supplement to this comment:
http://www.solid-earth-discuss.net/se-2016-154/se-2016-154-AC2-supplement.pdf